# Hymenoptera Venom Immunotherapy in Dogs: Safety and Clinical Efficacy

**DOI:** 10.3390/ani13193002

**Published:** 2023-09-23

**Authors:** Ana Rostaher, Nina Maria Fischer, Alessio Vigani, Barbara Steblaj, Franco Martini, Salina Brem, Claude Favrot, Mitja Kosnik

**Affiliations:** 1Dermatology Unit, Clinic for Small Animal Internal Medicine, Vetsuisse Faculty, University of Zurich, 8057 Zurich, Switzerland; nfischer@vetclinics.uzh.ch (N.M.F.); fmartini@vetclinics.uzh.ch (F.M.); sbrem@vetclinics.uzh.ch (S.B.); cfavrot@vetclinics.uzh.ch (C.F.); 2Division of Small Animal Emergency and Critical Care, Clinic for Small Animal Internal Medicine, Vetsuisse Faculty, University of Zurich, 8057 Zurich, Switzerland; avigani@vetclinics.uzh.ch; 3Section of Anaesthesiology, Department of Clinical Diagnostics and Services, Vetsuisse Faculty, University of Zurich, 8057 Zurich, Switzerland; bsteblaj@vetclinics.uzh.ch; 4Division of Allergy, University Clinic of Respiratory and Allergic Diseases Golnik, 4204 Golnik, Slovenia; mitja.kosnik@klinik-golnik.si; 5Faculty of Medicine, University of Ljubljana, 1000 Ljubljana, Slovenia

**Keywords:** anaphylaxis, angioedema, dogs, Hymenoptera allergy, urticaria, venom immunotherapy

## Abstract

**Simple Summary:**

Insect venom allergy is a potentially life-threatening allergic reaction following a bee, wasp, or ant sting. The only treatment to prevent further systemic sting reactions is venom immunotherapy (VIT), with an efficacy of up to 98% in humans. Prospective clinical data on VIT efficacy in dogs are currently lacking. In this investigation, 10 dogs with severe allergic reactions to either bee or wasp stings were treated with VIT. All dogs tolerated the therapy without adverse effects and the dogs which were re-stung tolerated the sting. This means that VIT is not only safe, but also efficacious in these patients. Furthermore, it was also shown that in addition to skin testing, two serum allergen-specific IgE tests were reliable to identify the underlying patients’ insect sensitization pattern.

**Abstract:**

Hymenoptera allergens are the main triggers for anaphylaxis in susceptible dogs and humans. Hymenoptera venom specific immunotherapy (VIT), the only disease-modifying treatment, has the potential to prevent future life-threatening reactions in human patients. Prospective clinical data on VIT efficacy in dogs are currently lacking. Therefore, the aim of this study was to show that VIT is not only safe but also efficacious in preventing anaphylaxis in dogs allergic to Hymenoptera. This uncontrolled prospective clinical trial included 10 client-owned dogs with a history of anaphylaxis following repeated Hymenoptera stings. The sensitization to bee and wasp allergens was demonstrated by intradermal testing (IDT) and allergen-specific IgE serology. For VIT induction (induction phase), dogs received a shortened rush immunotherapy protocol with aqueous allergens, which was then followed by monthly injections of 100 µg of alum-precipitated allergen (maintenance phase). VIT efficacy was determined by observing patients’ clinical reactions to re-stings. No systemic adverse events were seen during the induction and maintenance phases. From the seven re-stung dogs, only one developed a mild angioedema at the site of the sting; the remaining dogs were asymptomatic. These results show that VIT represents a safe and effective treatment option for Hymenoptera-allergic dogs.

## 1. Introduction

Anaphylaxis is a serious, life-threatening systemic hypersensitivity reaction characterized by a rapid onset, and manifests primarily with skin, airway, and circulatory changes. In humans, Hymenoptera venom (bee, wasp, ant) is recognized as one of the most frequent triggers of anaphylaxis with a prevalence ranging from 0.3 to 7.5% [1,2]. The same holds true for the dog [3,4]. The reported mortality rate is 0.5 per 1 million humans per year [5,6], but it is very likely that the mortality has been underestimated due to unrecognized stings in unexplained causes of death. It is estimated that 40 to 100 fatal sting reactions occur each year in the USA [7]. Human patients with a severe systemic allergic reaction to Hymenoptera stings have a 30% to 67% risk for recurrent anaphylaxis on a subsequent sting and thus need preventive measures [8,9,10].

In humans, venom immunotherapy (VIT) is the only specific treatment to prevent anaphylaxis after repeated allergen exposure [11]. Before VIT initiation, the sensitization to the offending allergen is documented by allergen-specific IgE serology and/or skin testing [12]. VIT is efficacious in 85% to 100% of patients for bee and wasp venom, respectively. A maximum dose of 100 μg venom allergen dose subcutaneously usually offers adequate protection in the majority of venom-allergic individuals. This dose is equivalent to the dry weight of approximately two bee stings or five wasp stings [13]. In humans undergoing VIT, treatment side effects, ranging from mild to severe allergic reactions, are observed in 8% to 20% of patients, primarily during the induction phase, when the desired maintenance dosage is reached by administration of small dose increments over time [8,11]. Therefore, the patients are strictly hospitalized during the induction phase. In conventional induction protocols, the time required to reach the maintenance dose is several weeks: in faster protocols such as rush and ultra-rush induction protocols, maintenance is achieved in several days or even hours, respectively. Fast induction has the obvious advantage of achieving rapid protection and is more convenient for the patient due to reduction in hospitalization time. It was shown that protection is very likely achieved immediately after reaching the maintenance dose [14]. VIT termination after one or two years leads to a relapse rate of 25%, and after 5 years less than 10% [15,16]. Therefore, the recommended duration for VIT in humans is 5 years or even life-long in patients with high risk of future bee stings [14,15]. In dogs, it was shown that VIT was well-tolerated in conventional and rush protocols, but prospective controlled trials on the efficacy are missing [3,17,18,19,20,21,22].

In this study, we aimed to determine the safety and efficacy of a VIT protocol consisting of a 160-min rush induction phase in dogs susceptible to anaphylaxis following Hymenoptera stings. We hypothesized that this VIT protocol would be well tolerated, and that at least 80% of the patients would be protected.

## 2. Materials and Methods

### 2.1. Study Population

This uncontrolled prospective clinical trial included 10 patients from the Vetsuisse Faculty Animal Hospital, University of Zurich, recruited between years 2022 and 2023. The following inclusion criteria were applied:Moderate or severe anaphylactic reaction, following an observed or suspected bee and/or wasp sting (grading system is shown in Table 1).A positive bee- and/or wasp-specific IgE serology or skin test.

Key exclusion criteria included pregnancy and any systemic, neoplastic or autoimmune disease. The dogs could be withdrawn from the study at any time when either welfare or health conditions were compromised (e.g., severe VIT adverse events), on pet owner’s request, or due to lack of compliance.

### 2.2. Anaphylaxis Grading Scale

The severity of each dog’s adverse reaction to the initial Hymenoptera sting was defined using a three-point (mild, moderate, or severe) grading scale, according to a recent published grading system [22].

### 2.3. Allergy Testing

Allergen-specific IgE levels to bee (*Apis mellifera*; Api m) and wasp allergens (*Vespula vulgaris*; Ves v) were measured by Allerceptᵀᴹ (Heska AG, Fribourg, Switzerland), an ELISA-based test utilizing whole allergen extracts. The results were expressed in Heska Epsilon Receptor Binding Units (HERBU) with a positivity threshold of 11 HERBU/mL for bee and wasp allergens [23]. In addition to whole allergen extracts, IgE levels to individual allergen components (nApi m 1, rApi m 10, rApi m 2, rApi m 3, rApi m 5, rVes v 1, rVes v 5) were measured by Pet Allergy Xplorerᵀᴹ (Nextmune, Lelystad, The Netherlands), with a positivity threshold of 28 ng/mL. Both tests utilized carbohydrate cross-reacting determinant (CCD)-blockers.

The skin test protocol used in this study was based on established human protocols and a recent protocol published for dogs [12,21]. Briefly, different allergen concentrations were applied to the skin via both the skin prick (SPT) and intradermal (IDT) tests [11]. Bee and wasp allergen extracts (Venomil, Bencard Allergie GmbH, Munich, Germany and Allergen Extrakte Dr. Weyers, Labor Dr. Weyers, Aachen, Germany) were administered by skin pricking using a lancet (Stallerpoint, Stallergenes S.A., Antony, France) and intradermal injections using BD Micro-Fine 0.3 mL insulin syringes (Becton Dickinson, Allschwil, Switzerland). All test solutions were applied on the patient’s left lateral thorax, and 15 min later, the test reactions were evaluated subjectively based on erythema and wheal size formation, as described previously [24]. A positive SPT and IDT were defined as wheal formation and an erythema score of ≥2 on a 0–4 subjective grading scale (Figure 1).

### 2.4. Allergen Selection for VIT

The allergen selection was based on the patient’s history (direct identification of the offending insect), and the allergy test results following human guidelines [11]. For monosensitized patients with a history of sting reactions to one insect species, VIT with one venom was performed. For patients experiencing anaphylaxis after being stung with more than one Hymenoptera species, or with severe initial reactions to one unidentified insect and equivalent positive reactions to bee and wasp venom in allergy tests, VIT with bee and wasp venoms was performed.

### 2.5. Protocol(s) for VIT

The induction immunotherapy protocol is described in detail in Table 2. Briefly, the dogs received subcutaneous bee and/or wasp aqueous extracts (Venomil, Bencard, Munich, Germany or Allergen Extrakte Dr. Weyers) by a standard rush induction protocol on days 1 and 7. The lyophilized extracts were kept at −20 °C until use. Extracts were reconstituted with sterile 0.5% phenol-buffered saline (Stallergenes, Dietlikon, Switzerland) to a final concentration of 100 µg/mL, rather than human serum albumin, to avoid a potential adverse immunological reaction to xenogens, as previously recommended [17,20,21]. For the 1st and 2nd VIT injection, the 100 µg/mL allergen solution was further diluted by 10 (1 µg dose) and 100 times (0.1 µg dose) to enable achieving the appropriate injection volume. On the first day, the dogs received a cumulative allergen dose of 101.1 µg divided into six subcutaneous injections given at 20-min intervals. Six days later, the dogs received an allergen dose of 100 µg divided into 2 injections at a 20-min interval. During the induction phase, the first seven dogs were premedicated with antihistamine Cetirizine (1 mg/kg daily; Cetirizin, Streuli Pharma, Uznach, Switzerland) starting three days before immunotherapy. In order to gain real-world data on the effect of Cetirizine on the occurrence of side effects during VIT, the last three dogs did not receive any premedication.

Thereafter, the dogs received maintenance therapy once monthly, consisting of 100 µg of alum-adjuvated venom allergen subcutaneously (Alutard, ALK, Wallisellen, Switzerland). After one year, the maintenance dosing interval was increased to six weeks, in third year to every eight weeks, in fourth year to every 10 weeks, and after 5th year every 12 weeks, as previously described [11]. The owners of the responder dogs were advised to continue the immunotherapy for the next five years or even life-long, following human VIT guidelines [11].

In case of severe adverse events during immunotherapy (moderate to severe anaphylaxis, see Table 1), VIT would be discontinued. In these circumstances, VIT would have been re-instituted within one week with the second last well-tolerated dose, as proposed in human VIT guidelines [11]. If adverse events prevented reaching the 100 µg maintenance dose, VIT would have been discontinued.

### 2.6. Safety and Efficacy Evaluation

During the induction phase, the dogs were continuously monitored for possible side effects with full emergency resuscitation equipment available if needed. The clinical parameters assessed for each dog included demeanor, mucous membrane color, capillary refill time, presence and severity of pruritus (with a three-point grading scale, ranging from mild, moderate to severe) and skin lesions (urticaria, angioedema, erythema) and continuously monitored (Life Scope^®^ BSM-6501K, Nihon Kohden, Rosbach, Germany) heart and respiratory rate, pulse rate and quality, body temperature, blood pressure, peripheral oxygen saturation. The adverse reactions were classified as absent, mild, moderate, or severe systemic reactions (Table 1).

In all patients, an intravenous (IV) catheter was placed, to be used in case of side effects. The treatment of side effects was tailored to the individual patient by current guidelines [25], and consisted of intravenous Ringer’s solution (10 mL/kg in 5–10 min, with a maximal volume 80–90 mL/kg/h), adrenaline (1 µg/kg every 1–2 min IV until mean arterial blood pressure is >60 mmHg), glucocorticoids (methylprednisolone, 1 mg/kg IV, Solu-MEDROL, Pfizer, Zurich, Switzerland) and H1-antihistamines (Clemastine, 0.05 mg/kg IV, Tavegyl, GlaxoSmithKline, Baar, Switzerland).

The efficacy was determined either by accidentally occurring stings, or an in-clinic challenge after 6 months (Figure 2), as published previously for humans undergoing VIT [26]. For the in-clinic challenge, the insect used was entomologically identified. Only bees which were already able to leave the nest were used (preferentially guardians at the entrance). Wasps were collected in nature close to their feeding sources with the help of attraction with foods. Insect handling during insect transportation or storage can lead to venom loss, which was reduced as much as possible. Insects were used within 12 h.

### 2.7. Endpoints and Statistics

The primary safety endpoint was assessed by determining the percentage of patients experiencing adverse effects during VIT. The clinical efficacy endpoint was determined by the percentage of patients tolerating a re-sting after an in-clinic or field provocation. Further numerical (weight, age, VIT duration) and categorical data (anaphylaxis grade, number of stings and anaphylaxis events, patients positive in SPT, IDT, and serological tests) were analyzed with descriptive statistics. The correlation between the PAX^®^ (Nextmune, Lelystad, The Netherlands) and Allercept^®^ (Heska AG, Fribourg, Switzerland) serological tests was calculated by Pearson’s correlation coefficient (r). The bee- and wasp-specific IgE levels were compared by Mann–Whitney test. Significance was set to *p*-values less than 0.05. The statistical analysis was carried out using the GraphPad Prism V8 (San Diego, CA, USA).

## 3. Results

### 3.1. Demographic and Clinical Data

The clinical data of the 10 enrolled patients are given in Table 3. Three male and seven female dogs with ages ranging from 0.8–8 years (median 2.6 years) were included. The median weight was 12.2 kg (range 3–31.2 kg). Four small-breed dogs (range 1–10 kg), three medium-sized dogs (range 10–25 kg), and three large-breed dogs (>25 kg) were included in this study. Most of the dogs were pure-bred dogs (9 out of 10); one was a mixed-breed dog. Nine out of ten dogs experienced a grade three anaphylactic reaction prior to VIT. Most of the dogs were exposed to two stings during their lifetime (range 1–3), as reported by the owners. Six out of ten dogs had a history of allergic reactions to previous insect stings; in five of these six dogs, the reaction severity worsened over time. The median age at the time of the first anaphylactic reaction was 1.5 years (range 0.6–7 years). Clinical manifestations of anaphylaxis included vomiting and/or diarrhea (6), collapse (8), urticaria or angioedema (3), and dyspnea (1).

### 3.2. Data from In Vitro and In Vivo Allergic Testing

Table 4 shows all data related to the allergological work-up. As reported by the owners, 8 out of 10 dogs had a history of allergic reactions to bee stings, one dog was stung by a wasp, and one dog had a history of bee and wasp stings.

All 10 dogs tested positive in IgE Allercept^®^ and 9 out of 10 dogs in the PAX^®^ serology test. When correlating the data of both serological tests, a statistically significant moderate positive correlation was found (r = 0.62, *p* = 0.028; Figure 3) for bee allergens. Eighty percent of the dogs (8/10 dogs) tested positive for CCD IgE, as determined by Allercept^®^ test, only dogs Number 5 and 9 were negative for these specific antibodies. The levels of bee-specific IgEs were significantly higher than the wasp-specific IgE for Allercept^®^ (median 350 vs. 30 HERBU/mL, *p* = 0.007) and PAX test (median 182 vs. 27 ng/mL, *p* = 0.003).

Most of the bee-allergic dogs were sensitized to the following allergen components: Api m 1, Api m 2, Api m 3, and Api m 10. None of the dogs were sensitized to Api m 5 (Table 5). As there were only two dogs allergic to wasps, the dataset for the allergen components was too small for statistical analysis.

IDT was positive in 9 out of 10 dogs and SPT in 3 dogs. All allergens were injected simultaneously, and no adverse events were observed. Nine dogs reacted in IDT to the 1 µg/mL allergen concentration, five reacted to 0.1 µg/mL, and four dogs reacted to the 0.01 µg/mL concentration.

Nine dogs with a history of developing anaphylaxis to only one insect species appeared to be sensitized to both bee and wasp allergens, when all test modalities were considered. Eight, three, and two out of nine were double-positive in Allercept^®^, IDT, and PAX^®^ testing, respectively.

### 3.3. Data on Venom Immunotherapy Outcomes

Table 6 shows the summary for the data regarding VIT. Based on the history and the allergological work-up, the allergens utilized in VIT were as follows: bee allergen in eight dogs, wasp in one dog, and bee and wasp allergens in one dog. Ten out of ten dogs completed the rush induction VIT phase and reached the maintenance dose of 100 µg allergen with no significant adverse reactions. Only two dogs developed a mild transient pruritus a few minutes after the 4th VIT injection (60 min after VIT initiation). Neither of the two had been premedicated with H1-antihistamine cetirizine.

No adverse events were observed during the maintenance phase with a median observation time of 343 days (90–702 days). Seven out of ten dogs were subjected to re-stings (three had an in-clinic provocation, four were stung accidentally). The median time from VIT induction to the re-sting was 182 days (range 12–360 days). Only one out of seven developed a mild angioedema at the sting area, which resolved within 24 h after receiving one oral dose of cetirizine (1 mg/kg) and prednisolone (1 mg/kg Hedylon, Graeub, Bern, Switzerland). Three owners refused a sting provocation and also did not report new field stings. They continued to carry the emergency medication (adrenaline, prednisolone, antihistamine) permanently with them.

## 4. Discussion

The only therapeutic options for life-threatening venom allergy are the prescription of adrenaline/epinephrine auto-injectors or VIT. VIT is currently the only treatment that specifically addresses the cause for such reactions. This is the first published prospective clinical trial on the efficacy of VIT in dogs, confirming efficacy similar to that for humans [11] and previous anecdotal or retrospective reports in dogs [17,19,20,24]. In our study, none of the re-stung dogs developed systemic allergic reactions. Only one dog developed a local reaction on the stinging area. Prior to VIT, this dog had developed a grade three anaphylactic reaction to a bee sting, meaning that the reaction severity after VIT diminished substantially. The timing of the re-sting differed from case to case, but most of the patients were stung within one year. Two dogs were accidentally stung immediately after the induction phase (12 and 30 days later) and both tolerated the sting. As shown in humans, the tolerance to Hymenoptera venom by VIT is quickly established, and almost 90% of patients are already protected against Hymenoptera stings one week after reaching the maintenance dose [14]. This early induction of tolerance is likely induced by silencing of mast cells and basophils [27].

Experiencing an anaphylactic reaction after an insect sting can have broad consequences for human patients. Not only is the event life-threatening, it can have a lasting influence on their quality of life (QoL) through the frightening experience [28]. The same can be assumed also for dogs and their caregivers. Given the effort required to avoid accidental exposures and the inherent uncertainty of VIT success, living with Hymenoptera venom allergy negatively influences the QoL. Although VIT significantly improves QoL in human patients [29], if such patients are offered a sting challenge and their reaction is minimal, their QoL after tolerating a sting is even higher [30,31]. In contrast, therapy with adrenaline autoinjector alone was shown to significantly increase the burden for patients [32] and is associated with a higher level of anxiety and depression [33]. In humans, several immunological changes have been associated with successful VIT. These include significant increases in circulating regulatory T cells and venom-specific IgG antibodies (particularly of the IgG4 subclass), decrease in basophil responsiveness against the venom allergens, and significant changes in secreted cytokines [11,34,35]. Despite these findings, up to date no biomarker for the evaluation of VIT efficacy has been established. Therefore, a controlled insect sting challenge remains the golden standard for the evaluation of venom tolerance, indicating clinical protection in humans [11,36,37]. In this study, 3 out of 10 dog owners refused a controlled sting challenge, which is in line with human data [14]. The owners of these dogs did not report on new field stings. It can be only speculated if the dogs had tolerated an unrecognized sting, or these owners were very cautious in avoiding contact with these insects.

The frequency of systemic adverse events with VIT ranges from 8% to 20% in humans [8,11] and from 35% to 47% in dogs [20,22]. These symptoms are mostly mild and adequately respond to standard anti-allergic treatment. In the case of systemic adverse events during immunotherapy, the allergen dose is reduced (going one to two steps back in the protocol), and premedication with H1-antihistamines should be used as it is associated with less side effects and possibly higher therapeutic success in VIT [11,38]. In this study, all dogs tolerated the induction and maintenance phases. Only two dogs developed mild pruritus at the injection site during the last injections in the build-up phase, which was seen only in dogs without premedication with H1-antihistamine Cetirizine. Based on these data, it seems that such premedication may be recommended to avoid skin-related side effects, but it cannot be used as a prophylactic measure to prevent systemic anaphylaxis, as shown in humans [11,39]. The higher adverse events rate in previous canine and human VIT studies as compared to our study can be explained by the fact that these studies used human serum albumin alongside the allergen extracts, which may act as a sensitizer and induce anaphylaxis [17,20,40].

Unambiguous identification of the clinically relevant venom is a prerequisite for optimal efficacy of VIT and adequate patient management. Thus, proving sensitization to a certain venom by skin testing and/or specific IgE (sIgE) measurements is imperative for the initiation of potentially life-saving VIT [11]. In this study, all Hymenoptera-specific IgE sensitizations could be identified by both IgE serology testing, and in 9 out of 10 dogs in the IDT. In the dog with negative reactions in IDT, the test was performed 10 days after the anaphylactic event and may represent a false negative result during the refractory period [12]. For this reason, it is recommended to perform the IDT at least two weeks after such episodes [12]. Sensitization to the venoms of more than one Hymenoptera species is common in insect venom-allergic patients [41,42] and was also observed with our canine patients. In case of an unclear clinical history, it can be difficult to determine whether this reflects double sensitization due to cross-reactivity of shared allergenic determinants or genuine multiple sensitizations to more than one venom. However, in most of these cases, treatment with only one venom appears to be sufficient [41]. A major diagnostic problem is that currently available tests, such as serum allergen-specific IgE (including assessment of molecular allergens), intradermal or skin prick testing, and/or the basophil-activation tests, cannot distinguish between asymptomatic sensitization and clinically relevant Hymenoptera allergy [43]. The current human VIT guidelines therefore recommend that VIT with double venoms (bee and wasp) is indicated in clinically allergic patients reporting systemic reactions to stings of both Hymenoptera spp., and in those with equal reactivity to both venoms in diagnostic tests, when these patients have not reliably identified the culprit stinging insect [11]. It should also be emphasized that the results of skin and serological tests cannot be used as a diagnostic tool to identify Hymenoptera venom allergy [11] and they also do not correlate with the severity of symptoms after the sting [44]. The prevalence of sensitization (indicated by a positive skin test or the detection of venom-specific IgE in the serum) is estimated at between 9.3% and 28.7% in the healthy human adult population [5]. The prevalence of being stung by Hymenoptera species during life ranges from 56.6% to 94.5% in adults and 37.5% in children [45], which may explain the high sensitization prevalence in the healthy population. Such epidemiological data are currently missing in veterinary medicine, but it can be assumed that the same holds true for dogs, as they are exposed to the same environment and environmental risks.

In our study, the clinical history was helpful in discerning the clinically relevant sensitization, as 9 and 3 out of 10 patients were polysensitized to both bee and wasp allergens, as determined by serology and IDT, respectively. Very likely these patients were stung previously with these insects without clinical relevance. Furthermore, in this study, a simultaneous IDT with several allergen concentrations was performed. The results showed that most of the dogs were positive at the concentration of 1 µg/mL. This is in line with human studies showing that a simultaneous multi-step [46] or even single-step IDT [46] at a concentration of 1 µg/mL is safe, faster, and more cost-effective compared with the standard sequential testing approach [8].

Common allergy-relevant species of Hymenoptera include bees and yellow jackets, which are found all over the world, but particularly in the northern hemisphere; these and members of the genus Polistes are of great importance in the USA and Southern Europe [47]. In our study, the most prevalent allergen was the bee. Only two patients reacted to wasps. If a patient shows clinical reactivity to bumble bee, for which there are no commercially available extracts, bee extracts can be used, as cross-reactivity is common among bee spp. [11]. In Europe, VIT may be performed either by aqueous extracts or depot preparations [11]. The aqueous preparations are used in ultra-rush, rush, and clustered induction protocols, whereas purified aluminum hydroxide adsorbed preparations are typically used only in the conventional and cluster induction protocols and in the maintenance phases [11]. Treatment can be switched from aqueous to depot preparations following the rapid up-dosing phase [48]; this was the case in this study, without any observed side effects.

Venom immunotherapy is a dynamic treatment in which individuals with an allergy are administered a specific allergen to desensitize and modulate mechanisms of their allergic hypersensitivity over time. VIT can be given as ultra-rush, rush, cluster protocols, and conventional protocols with multiple venom exposures within hours, days, weeks, or months, respectively. All of these methods are proven to be safe and effective, although the slower protocols may be associated with higher rates of venom immunotherapy-induced anaphylaxis [49,50] and even lower efficacy [51]. However, contrasting reports exist [52]. Adverse events tend to occur more frequently during early dosing phases, and decrease in both incidence and severity during maintenance phase. The advantage of accelerated VIT protocols, such as the one reported in this study, is that the protection is achieved rapidly without increasing the risk for systemic side effects. Additionally, an abbreviated VIT induction phase may also improve client compliance due to time and cost (less visits) savings. From human guidelines, the recommended starting dose in up-dosing protocols lies between 0.001 and 0.1 µg, but it has also been shown that a starting dose of 1 µg is safe and not associated with a higher rate of side effects in adults or in children [11,53]. In our study, the starting dose was 0.1 µg and was tolerated in all dogs. In order to shorten the therapy duration, a starting dose of 1 µg could be explored in the future. In a recent study evaluating the safety of a modified rush VIT protocol in dogs, the starting dose was 0.05 µg, and the patients received in total 14 injections on 3 different days [22], which is slightly more than in our study. In this study, after one year of monthly injections, an extended maintenance dose regime was initiated. It was shown in humans that it is as effective and safe as the conventional monthly dosing and seems to be the best option in terms of convenience and economic savings [54].

Although subcutaneous VIT with a standard dose of 100 μg is a highly effective treatment, some human patients, as identified by sting challenge, develop tolerance only if the VIT allergen dose is increased 200 µg [55]. This is also why a sting challenge should be performed, whenever possible. Despite the high success of VIT, allergen tolerance may not persist for a prolonged time after stopping treatment. Most of the studies concluded that a minimum of a five-year treatment is superior for long-term effectiveness [15,56,57,58]. Anaphylaxis relapse rates of 0–10% and 17%, one to five years after stopping vespid and 1 year for bee VIT, have been reported [11]. Life-long therapy should be considered in patients with severe initial systemic reactions, systemic adverse events during VIT, and bee venom-allergic patients with high risk of future bee stings [11]. As dogs usually are at high risk to encounter new stings, the authors recommend life-long VIT therapy in most of the dogs.

The main limitation of this study is that a placebo group was not included, as the authors wanted to avoid an unnecessary exposure of placebo-treated patients to life-threatening reactions. The likelihood for recurrent anaphylaxis can be approximately 60% in placebo-treated humans [11] and similarly 63% of naturally immunized dogs with two stings are prone to recurrent anaphylaxis (A.R, unpublished data). Therefore, we speculate that at least 60% of the dogs in this study were not subjected to a placebo-effect and actually profited from VIT.

## 5. Conclusions

This investigation showed that a two-day rush VIT protocol of a 160-min duration was safe and efficient in dogs experiencing severe anaphylaxis to bee and/or wasp stings. The two allergen-specific IgE testing methods and IDT were all reliable in identifying the underlying Hymenoptera sensitizations. Furthermore, it was shown that the sensitivity of IDT was the highest at 1 µg/mL allergen concentration without any side effects. Future large-scale studies should confirm these results and also the long-term sustained clinical efficacy after discontinuation of VIT in dogs.

## Figures and Tables

**Figure 1 animals-13-03002-f001:**
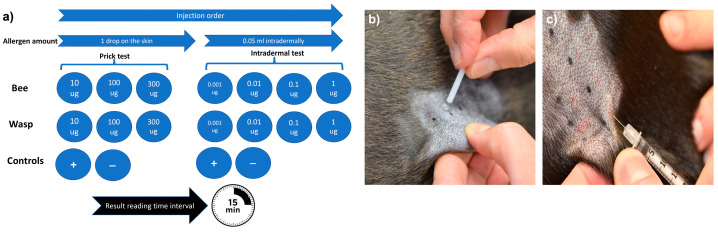
Skin testing procedure. The sequence and allergen concentrations for the skin testing (**a**), skin prick testing procedure (**b**), and intradermal testing procedure (**c**). All allergens were injected simultaneously.

**Figure 2 animals-13-03002-f002:**
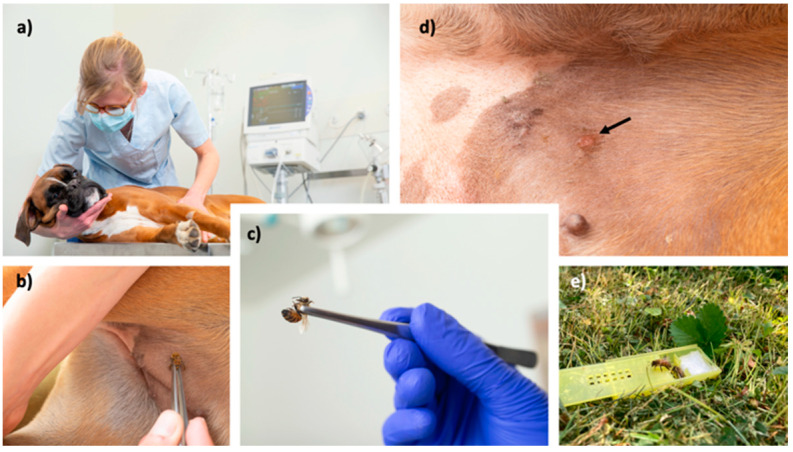
Presentation of the in-clinic challenge with a bee. Patients are closely monitored during in-clinic challenge (**a**), the insect is gently grabbed by forceps and brought to the skin in the inguinal area (**b**,**c**). In case of bee stings, their stinger is removed after a few minutes. At the stung area, a well-defined erythematous wheal (arrow) should appear (**d**). If no reaction is observed, the procedure is repeated with a new insect. Finally, the remaining insects are transported back and freed into their natural environment (**e**).

**Figure 3 animals-13-03002-f003:**
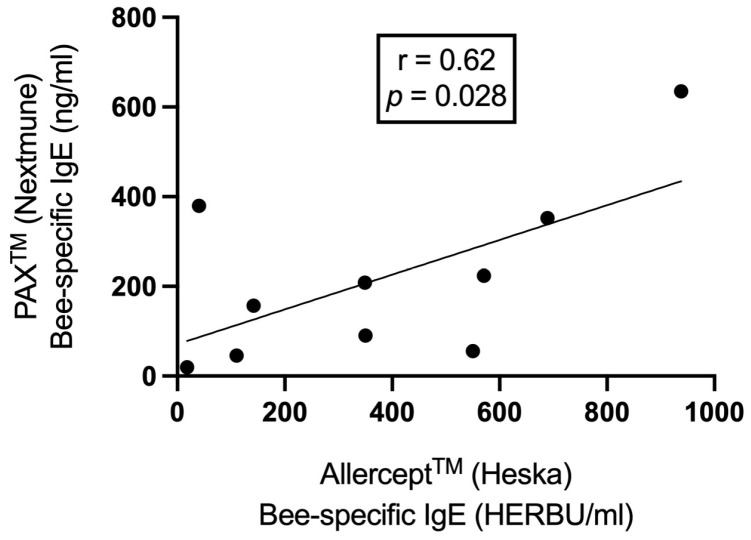
The correlation between PAX^®^ and Allercept^®^-derived bee-specific IgE levels by Pearson’s correlation coefficient.

**Table 1 animals-13-03002-t001:** Clinical grading system for anaphylaxis (adapted from Moore et al. [22]).

Grade	Organ System Involved	Clinical Findings
**1—mild**	Skin	Generalized erythema, urticaria and/or angioedema
**2—moderate**	Gastrointestinal, respiratory ± skin	Dyspnea, stridor, wheeze, nausea, vomiting and/or abdominal pain ± above findings
**3—severe**	Respiratory, cardiovascular, neurological ± skin ± gastrointestinal, respiratory	Cyanosis, pallor, SpO_2_ < 92%, hypotension (systolic blood pressure < 90 mmHg), collapse, loss of consciousness and/or incontinence ± above findings

**Table 2 animals-13-03002-t002:** Induction protocol with information on allergen concentration, amount, volume, and observation time.

	Injection No.	Allergen Amount (µg)	Concentration (µg/mL)	Injection Volume (mL)	Observation Time (Minutes)
Day 1	1	0.1	1	0.1	20
	2	1	10	0.1	20
	3	10	100	0.1	20
	4	20	100	0.2	20
	5	30	100	0.3	20
	6	40	100	0.4	20
Day 7	7	50	100	0.5	20
	8	50	100	0.5	20

**Table 3 animals-13-03002-t003:** Clinical characteristics of the patient population. Legend: N/A, not applicable; reaction severity ranges on scale from 1–3 (mild–severe).

Signalment at VIT Initiation						Clinical Data Related to the Sting Event					
Dog N^o^	Breed	Age (Years)	Sex	Castration Status	Weight (kg)	No. of Sting Episodes	No. of Anaphylaxis Episodes	Age 1st Sting (Years)	Age 1st Anaphylaxis Episode (Years)	Reaction Severity (1st Sting)	Reaction Severity (Last Sting)
1	Boxer	3.0	female	intact	28.6	2	2	0.75	2.8	2	3
2	Toy poodle	2.0	male	intact	3	1	1	1.75	1.75	3	N/A
3	Dachshund	4.0	female	spayed	5.8	2	2	3.9	4	2	3
4	Dachshund	1.0	female	spayed	5.2	2	2	0.75	0.8	2	3
5	Entlebucher Sennenhund	1.0	female	spayed	18.3	1	1	1	1	3	N/A
6	Yorkshire Terrier	2.2	female	spayed	4.1	2	2	1	1.2	2	3
7	Cross-breed	7.4	male	castrated	10.5	1	1	7	7	3	N/A
8	Malinois	3.1	female	spayed	25.1	3	2	1.1	1.3	1	2
9	Doberman	0.8	male	intact	31.2	1	1	0.6	0.6	3	N/A
10	Beagle	8.0	female	spayed	13.9	3	3	4	4	3	3

**Table 4 animals-13-03002-t004:** Data on the results of the allergological work-up. First, results of the prick (SPT) and intradermal testing (IDT) are shown. Additionally, the serum level of bee- and wasp-specific IgE, the time interval between the anaphylactic event and the testing, and owner insect identification are shown. The positive test results are depicted in blue (bee allergen), and gray (wasp allergen) shadow and the cut-off values were 11 HERBU/mL for Allercept and 28 ng/mL for PAX. *—Dog No. 7 was PAX positive for the wasp allergen component rVes v 5, although not for the wasp allergen extract and is therefore marked as positive in gray shadow.

Dog No.	Bee Allergens				Wasp Allergens				Insect Identification by Owner	Time Interval Sting to Testing
	Prick	IDT	Serology		Prick	IDT	Serology			
	Positive at Following Concentrations (µg/mL)		Allercept (HERBU/mL)	PAX (ng/mL)	Positive at Following Concentrations (µg/mL)		Allercept (HERBU/mL)	PAX (ng/mL)		
1	Neg.	1	142	156.67	Neg.	Neg.	116	28.02	Bee	4 weeks
2	100–300	1	350	90.54	Neg.	Neg.	84	27.2	Bee	3 weeks
3	Neg.	Neg.	938	634.7	Neg.	Neg.	45	23.62	Bee	10 days
4	Neg.	0.01–1	349	208.14	Neg.	0.01–1	16	32.22	Bee	5 weeks
5	Neg.	0.01–1	110	45.77	Neg.	0.01–1	16	27.82	Bee	6 weeks
6	10–300	1	40	379.62	Neg.	Neg.	2	25.26	Bee	1 year
7	300	0.1–1	550	55.95	Neg.	0.1–1	36	27.33 *	Bee	4 months
8	Neg.	0.01–1	689	352	Neg.	Neg.	23	27	Bee	2 months
9	Neg.	Neg.	18	19.81	Neg.	1	18	25.45	Wasp	4 weeks
10	Neg.	0.01–1	571	223.44	Neg.	1	427	66.02	Bee and wasp	1 year

**Table 5 animals-13-03002-t005:** Data on the presence and levels of IgE to different bee allergen components. The positive test results are depicted in blue shadow.

Dog No.	PAX Results in Bee-Allergic Dogs					
	Bee Extract	Bee Allergen Components				
	Api m	nApi m 1	rApi m 2	rApi m 3	rApi m 5	rApi m 10
1	156.67	227.91	82.79	31.33	20.66	577.49
2	90.54	132.83	633.33	110.07	19.11	633.33
3	634.7	722.22	111.15	804.65	18.3	1478.85
4	208.14	592.32	853.02	987.17	21.57	1061.81
5	45.77	42.55	33.8	24.04	19.35	167.46
6	379.62	958.41	20.35	82.96	19.52	23.18
7	55.95	55.95	57.91	128.53	18.81	133.62
8	352	386	344	60	22	1657
10	223.44	356	234.32	467.19	19.23	1018.54

**Table 6 animals-13-03002-t006:** Venom immunotherapy (VIT) data. This table shows the relevant data regarding the allergen source used for VIT, if premedication with H1-antihistamines was given, data on side effects during the induction and maintenance phase, efficacy, and duration of follow-up. The asterisk signifies that the allergen used was Venomil (Bencard), the rest of the patients received allergen extract Dr. Weyers (Labor Dr. Weyers).

Dog No.	VIT Allergen Source	Premedication with Cetirizine	Side Effects VIT		Re-Stings			VIT Follow-Up (Days)
			Induction Phase	Maintenance Phase	Re-Sting No.	Clinical Signs	VIT Duration at First Re-Sting (Days)	
1	Bee *	Yes	No	No	1	No	360	620
2	Bee	Yes	No	No	1	No	269	614
3	Bee	Yes	No	No	1	Local angioedema	176	375
4	Bee	Yes	No	No	0	Not stung	N/A	312
5	Bee	Yes	No	No	0	Not stung	N/A	312
6	Bee	Yes	No	No	1	No	182	409
7	Bee	Yes	No	No	0	Not stung	N/A	202
8	Bee	No	Mild pruritus	No	1	No	12	90
9	Wasp	No	Mild pruritus	No	1	No	30	291
10	Bee * and wasp	No	No	No	1	No	307	702

## Data Availability

All data necessary to replicate this analysis are contained within this article.

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
