# Peer review of "Hymenoptera Venom Immunotherapy in Dogs: Safety and Clinical Efficacy"

_animals, 2023, doi:10.3390/ani13193002_

Round 1

Reviewer 1 Report

The study is well structured and clearly presented. Unfortunately the number of cases included in the study is very low. You may explain why yo decided to publish the article with few cases (finding clinical cases, owners compliance, etc...).

English language is fluent. Minor revisions.

Author Response

Please find enclosed our revised manuscript, entitled „ Hymenoptera Venom Immunotherapy in dogs: safety and clinical efficacy”, which is intended for publication in Animals.

We would like to thank the editor and the reviewers for positive and constructive comments regarding our manuscript. All changes in the manuscript are indicated in bold.
The article was proof-read by a native speaker and veterinary immunologist Dr. Valerie Fadok, PhD, diplACVD.

General considerations

The study is well structured and clearly presented, even if the number of patients included in the study is quite

low. In the clinical practice, in how many cases might clients, considering that treatment should be constantly repeated throughout the patient’s life, require VIT?

Answer: We do VIT 5 cases per year and we could observe that most cases are seen by general practitioners, who still are not aware that VIT is a treatment option for such cases. I think there is a large unawareness in this field, even in our emergency department. Our emergency department sees around 50 Hymenoptera anaphylaxis cases per year. Also there is also a strong unawareness in the owners, that such preventive measure is possible and that it is efficient.

Minor revisions

Lines 148-154. May you specify in which volume did you dissolved the powder? In table 2, you reported the allergen concentration (μg) per unit of volume (ml). I see that it ranges from 1 μg/ml to 100μg/ml, so I suppose you’ve dissolved the allergen powder in different volumes. Is it right?

Answer: No, we did not put it in different volumes, the end concentration was always 100 ug/ml and this one was necessary to dilute further for the first two injections as the injected amount  would have been to small. We gave this information now in the text:

“For the 1st and 2nd VIT injection, the 100 mg/ml allergen solution was further diluted by 10 (1 mg dose) and 100 times (0.1 mg dose) to enable achieving the appropriate injection volume.”

Lines 154-156. On what criteria were some dogs treated with antihistamines and others not?

Answer: We do not write here any criteria for antihistamines as they were none. Also, in humans there are no guidelines if AH should be used, it is just known that they can prevent occurrence of skin side effects (erythema, redness, urticaria, angioedema); clearly, they cannot protect from anaphylaxis.

We used cetirizine in the first 7 cases. As we did not observe any side effects in these 7 patients and in order to see if it is actually necessary to be used, we decided also to try VIT without premedication. From our data we can say that cetirizine can alleviate pruritus during VIT. It cannot alleviate severe side effects, what is known from humans, so therefore some human allergologists do not use it as it may mask a beginning severe anaphylactic reaction.

We adapted the text accordingly and wrote the rationale for using cetirizine:

“During the induction phase, the first seven dogs were premedicated with antihistamine Cetirizine (1 mg/kg daily; Cetirizin, Streuli Pharma, Uznach, Switzerland) starting three days before immunotherapy. In order to gain real-world data on the effect of cetirizine on the occurrence of side effects during VIT, the last three dogs did not receive any premedication.”

As we find your comment very useful, we integrated it into the discussion, together with data from humans:
“Only two dogs developed mild pruritus at the injection site during the last injections in the build-up phase, which was seen only in dogs without premedication with H1-antihistamine Cetirizine. Based on this data it seems that such premedication may be recommended to avoid skin related side effects, but it cannot be used as a prophylactic measure to prevent systemic anaphylaxis, as shown in humans.11,42»

Lines 165-168. I suggest rewriting this sentence.

Answer: We rewrote the sentence.

“In case of severe adverse events during immunotherapy (moderate to severe anaphylaxis, see table 1), VIT would be discontinued. In these circumstances, VIT would have been re-instituted within one week with the second last well-tolerated dose, as proposed in human VIT guidelines.11 If adverse events prevented reaching the 100 mg maintenance dose, VIT would have been discontinued.”

Lines 189-190. I suggest describing the systemic reaction as “mild, moderate, or severe” instead of “grade 1,2 or 3.”

Answer: We changed this. We write now:

“The adverse reactions were classified as absent, mild, moderate or severe systemic reactions (Table 1).”

Figure 1. In Figure 1 caption, you generally wrote about “insects”, but I can see only bees. Did the protocol differ for wasps? Since wasps do not dead after stinging, may you explain how did you manage these insects? There was no different procedure.

Answer: Thank you good point. In Figure 1 we mention: Presentation of the in-clinic challenge with a bee. The only difference is in collecting the insects (described in the text). The wasps (we have few for reserve) are then put back in the yellow box (we hold them all the time with the forceps) and freed in the nature like with did with the remaining bees.

Lines 223-224. Consider revising the sentence. Here a suggestion (from Reverso) “The statistical analysis was carried out using the GraphPad Prism V8 (San Diego, CA, USA).”

Answer: we changed it thank you. We write now:

“The statistical analysis was carried out using the GraphPad Prism V8 (San Diego, CA, USA).”

Line 326. Consider revising the sentence. Here a suggestion (from Reverso) “Neither had been premedicated with H1-antihistamine cetirizine.”

Answer: we changed it thank you. We write now:

“Neither had been premedicated with H1-antihistamine cetirizine.”

Lines 361-362. May you describe the behaviors the dogs put in place after the anaphylactic reaction to bee or wasp stings, indicative of a state of fear against these insects?

Answer: Here we just wanted to describe that humans are afraid of new stings and also human owners are afraid. Dogs usually again like to play with insects. But studies on life-quality are also largely awaited.

Reviewer 2 Report

This is a well-written paper incorporating excellent and novel clinical techniques and high quality science. This reviewer has few comments:

(i). The use of ceterizine as a premedicant to VIT seems to be random, and was not employed in 3 cases - in two of which mild pruritus was noted. This deserves comment. Do the authors now recommend its use as a routine? 

(ii) The manuscript would be improved if the authors could supply data concerning the positive predictive value of the serological assays employed. In the absence of direct identification of the insect involved in an anaphylactic occurrence, workers wishing to utilize the protocols employed in this paper may lack access to intradermal or skin prick testing as an aid to identifying the implicated allergen(s) and thus rely on serology. Does data exist on the incidence of positive IgE serology to insects in the normal population that have avoided any known anaphylactic reaction? 

(iii). The final paragraph is acknowledged to be speculative, but but could be strengthened if some numbers were given, rather than saying "...similarly about 60% of naturally immunized dogs with two stings are prone to recurrent anaphylaxis"

(iv) Line 462 should read "whenever possible" rather than "whener possible"

Author Response

Please find enclosed our revised manuscript, entitled „ Hymenoptera Venom Immunotherapy in dogs: safety and clinical efficacy”, which is intended for publication in Animals.

We would like to thank the editor and the reviewers for positive and constructive comments regarding our manuscript. All changes in the manuscript are indicated in bold.
The article was proof-read by a native speaker and veterinary immunologist Dr. Valerie Fadok, PhD, diplACVD.

This is a well-written paper incorporating excellent and novel clinical techniques and high-quality science. This reviewer has few comments:

(i). The use of ceterizine as a premedicant to VIT seems to be random, and was not employed in 3 cases - in two of which mild pruritus was noted. This deserves comment. Do the authors now recommend its use as a routine?

Answer: We used cetirizine in the first 7 cases. As we did not observe any side effects in these 7 patients and in order to see if it is actually necessary to be used, we decided also to try VIT without premedication. From our data we can say that cetirizine can alleviate pruritus during VIT. It cannot alleviate severe side effects, what is known from humans, so therefore some human allergologists do not use it as it may mask a beginning severe anaphylactic reaction.

We adapted the text accordingly and wrote the rationale for using cetirizine:

“During the induction phase, the first seven dogs were premedicated with antihistamine Cetirizine (1 mg/kg daily; Cetirizin, Streuli Pharma, Uznach, Switzerland) starting three days before immunotherapy. In order to gain real-world data on the effect of Cetirizine on the occurrence of side effects during VIT, the last three dogs did not receive any premedication.”

We also added our recommendation and data from humans on this topic which is in the discussion:
“Only two dogs developed mild pruritus at the injection site during the last injections in the build-up phase, which was seen only in dogs without premedication with H1-antihistamine Cetirizine. Based on this data it seems that such premedication may be recommended to avoid skin related side effects, but it cannot be used as a prophylactic measure to prevent systemic anaphylaxis, as shown in humans.11,42»

(ii) The manuscript would be improved if the authors could supply data concerning the positive predictive value of the serological assays employed. In the absence of direct identification of the insect involved in an anaphylactic occurrence, workers wishing to utilize the protocols employed in this paper may lack access to intradermal or skin prick testing as an aid to identifying the implicated allergen(s) and thus rely on serology. Does data exist on the incidence of positive IgE serology to insects in the normal population that have avoided any known anaphylactic reaction? 

Answer: This is a very good point, which we now discuss in the discussion. In humans it is known that more than 50% of the general population are stung by Hymenoptera insects at least once in their life’s and therefore it is not surprising that in the healthy human population IgE sensitization can be as high as 28% (less in the north and more in the south, which parallels the frequency of occurrence of these insects – longer season in the south than in the north). We write now the following:

“It should be also emphasized that the results of skin and serological tests cannot be used as a diagnostic tool to identify Hymenoptera venom allergy11 and they also do not correlate with the severity of symptoms after the sting.47 The prevalence of sensitization (indicated by a positive skin test or the detection of venom specific IgE in the serum) is estimated at between 9.3% and 28.7% in the healthy human adult population.5 The prevalence of being stung by Hymenoptera species during life ranges from 56.6 to 94.5% in adults and 37.5% in children,48 which may explain the high sensitization prevalence in the healthy population. Such epidemiological data are currently missing in veterinary medicine, but it can be assumed that the same holds true for the dog, as they are exposed to the same environment and environmental risks. “

(iii). The final paragraph is acknowledged to be speculative, but but could be strengthened if some numbers were given, rather than saying "...similarly about 60% of naturally immunized dogs with two stings are prone to recurrent anaphylaxis"

Answer: We give now the precise number (63%). This data will hopefully soon be also published.

(iv) Line 462 should read "whenever possible" rather than "whener possible"

Answer: Thank you, this is changed.